# Microsatellite analysis reveals connectivity among geographically distant transmission zones of *Plasmodium vivax* in the Peruvian Amazon: A critical barrier to regional malaria elimination

Paulo Manrique[1]*, Julio Miranda-Alban[1], Jhonatan Alarcon-Baldeon[1], Roberson Ramirez[1], Gabriel Carrasco-Escobar[1,2], Henry Herrera[1], Mitchel Guzman-Guzman[1], Angel Rosas-Aguirre[2,3,4], Alejandro Llanos-Cuentas[2,5], Joseph M. Vinetz[1,2,6,7], Ananias A. Escalante[8], Dionicia Gamboa[1,2,7]

1 Laboratorio ICEMR-Amazonia, Laboratorios de Investigación y Desarrollo, Facultad de Ciencias y Filosofa, Universidad Peruana Cayetano Heredia, Lima, Perú, 2 Instituto de Medicina Tropical Alexander von Humboldt, Universidad Peruana Cayetano Heredia, Lima, Perú, 3 Fund for Scientific Research FNRS, Brussels, Belgium, 4 Research Institute of Health and Society (IRSS), Université catholique de Louvain, Brussels, Belgium, 5 Facultad de Salud Pública y Administración, Universidad Peruana Cayetano Heredia, Lima, Perú, 6 Yale School of Medicine, Section of Infectious Diseases, Department of Internal Medicine, New Haven, Connecticut, United States of America, 7 Departamento de Ciencias Celulares y Moleculares, Facultad de Ciencias y Filosofía, Universidad Peruana Cayetano Heredia, Lima, Perú, 8 Institute for Genomics and Evolutionary Medicine (IGEM), Temple University, Philadelphia, Pennsylvania, United States of America

* paulonvnv@gmail.com

## Abstract

Despite efforts made over decades by the Peruvian government to eliminate malaria, *Plasmodium vivax* remains a challenge for public health decision-makers in the country. The uneven distribution of its incidence, plus its complex pattern of dispersion, has made ineffective control measures based on global information that lack the necessary detail to understand transmission fully. In this sense, population genetic tools can complement current surveillance. This study describes the genetic diversity and population structure from September 2012 to March 2015 in three geographically distant settlements, Cahuide (CAH), Lupuna (LUP) and Santa Emilia (STE), located in the Peruvian Amazon. A total 777 *P. vivax* mono-infections, out of 3264, were genotyped. Among study areas, LUP showed 19.7% of polyclonal infections, and its genetic diversity ($H_{exp}$) was 0.544. Temporal analysis showed a significant increment of polyclonal infections and $H_{exp}$, and the introduction and persistence of a new parasite population since March 2013. In STE, 40.1% of infections were polyclonal, with $H_{exp}$ = 0.596. The presence of four genetic clusters without signals of clonal expansion and infections with lower parasite densities compared against the other two areas were also found. At least four parasite populations were present in CAH in 2012, where, after June 2014, malaria cases decreased from 213 to 61, concomitant with a decrease in polyclonal infections (from 0.286 to 0.18), and expectedly variable $H_{exp}$. Strong signals of gene flow were present in the study areas and wide geographic distribution of

**Data Availability Statement:** All relevant data are within the manuscript and its Supporting Information files.

**Funding:** This study was funded by National Institutes of Health-National Institute of Allergy and Infectious Diseases (NIH-NIAID) U19AI089681 to JMV (https://www.niaid.nih.gov); and Training Grant 5D43TW007120 (https://www.fic.nih.gov). The funders had no role in study design, data collection and analysis, decision to publish, or preparation of the manuscript.

**Competing interests:** The authors have declared that no competing interests exist.

highly diverse parasite populations were found. This study suggests that movement of malaria parasites by human reservoirs connects geographically distant malaria transmission areas in the Peruvian Amazon. The maintenance of high levels of parasite genetic diversity through human mobility is a critical barrier to malaria elimination in this region.

## Author summary

*Plasmodium vivax* transmission is heterogeneous and discontinuous in the Peruvian Amazon. Such heterogeneity is the result of factors that include, but are not restricted to, the environment, public policies, and characteristics of the parasite, the vector, and human activities. All these factors make *P. vivax* transmission resilient to interventions. In order to achieve the goals of control and local elimination, *P. vivax* surveillance must inform how those factors sustain disease transmission in order to focalize and synchronize control strategies. In this study, we implemented molecular surveillance complemented with population genetic tools in the areas of Cahuide, Lupuna, and Santa Emilia located in the Peruvian Amazon. In particular, we characterize the transmission and the parasite genetic variation in these sites from September 2012 to March 2015. The changes in parasite diversity, the wide geographic dispersion of parasite subpopulation and the introduction of a new parasite clone or subpopulation in Lupuna documented in this study suggest that connectivity among the different endemic areas, likely due to human mobility, sustains disease transmission in the region hindering the success of control measures. This information must be considered in the design of current control strategies.

## Introduction

*Plasmodium vivax* is responsible for about 80% of malaria cases in the Peruvian Amazon region [1]. Even though its incidence slightly decreased in the last 3 years; historically, malaria transmission in Amazonia has waxed and waned due to multiple factors [2, 3]. The effectiveness and sustainability of malaria control efforts are strongly influenced by climatological events (e.g., *El Niño* Southern Oscillation), changes in public policies [2], changes in vector behavior [3–7], the high proportion of infections that escape traditional interventions and surveillance (asymptomatic, submicroscopic, symptomatic persons that do not seek for attention, and relapses) [3, 8, 9], and the spatial heterogeneity and transmission of this parasite [10–13].

Epidemiological and population genetic data showed that *P. vivax* is distributed in discrete geographical units with different levels of transmission in Loreto Department in the Peruvian Amazon [3, 9–11, 13]. The population at risk for malaria is mainly distributed in rural and peri-urban areas in Loreto [3]. In these areas, the main transport over long distances is by the rivers. Economic and social activities that promote human movement drive people towards various crossroads and economic centers [10], which is thought to bridge among different transmission foci leading to parasite movement and new introductions. The connection among malaria transmission sites mediated by the mobility of infected individuals could counteract the effect of habitat fragmentation in parasite populations [14], thus enabling reintroduction to areas where it has already been controlled [12].

Parasite gene flow among endemic areas could also expose the human population to a wider antigen repertoire leading to overcoming clinical immunity with symptomatic acute malaria. As the acquisition of anti-parasite and anti-disease immunity (i.e., asymptomatic

parasitemia, premunition) may depend on the number of infections, in turn, related to age and exposure [15–18], this complicated pattern of distribution and transmission of *P. vivax* populations might also play a role in the acquisition of acquired immunity.

A 3-year longitudinal cohort study in two epidemiologically contrasting settings, San Jose de Lupuna (LUP) and Cahuide (CAH) was carried out from 2012 to 2015. Another one-year cohort was established in a remote malaria-endemic village, Santa Emilia (STE), in 2013. These areas are geographically distant from each other and river travel is the only communication channel from LUP and STE to other places; CAH is located along a road (the Iquitos-Nauta highway) as well as on the Itaya River [9]. Villagers in the three study sites mainly work in agricultural and other extracting activities like logging, fishing, and hunting; but people regularly travel to trade and barter their products [9]. Considering this context, little or no reduction in genetic diversity and signals of gene flow among parasite populations would be expected as a result of the local human mobility. In this investigation, we studied changes in genetic diversity, temporal replacement of populations, processes of importation, introduction, and clonal propagation of *P. vivax* in these study sites. Moreover, the replacement of parasite clones and changes in parasite density in individuals with recurrent episodes were analyzed as an approach of anti-parasite immunity acquisition.

## Methods

### Ethics statement

The study was approved by the Ethical Committee of Universidad Peruana Cayetano Heredia, Lima- Peru (SIDISI codes: 57395 and 60429) and the University of California San Diego Human Subjects Protection Program (Project # 100765). Permissions were received from health and local authorities after explaining the purpose and procedures of the study. Signed informed consent was obtained prior to the study enrollment to participation and blood sampling by all adults and the parents of all participating children <18 years. In addition to parental/guardian consent, children older than 7 years provided a signed informed assent. All the methods were carried out in accordance with approved guidelines.

### Study area and population

The study was conducted in Cahuide (CAH), Lupuna (LUP) and Santa Emilia (STE) sites in the Peruvian Amazon, Department of Loreto (Fig 1). CAH and LUP sites have been described previously [9]. LUP is a forested area near the administrative border between San Juan and Iquitos districts (latitude 03˚ 44.591' S longitude 73˚ 19.615' W), only accessible by crossing the Nanay River by boat (45 minutes trip) from Iquitos. In LUP individuals are distributed among its three adjacent villages Santa Rita (SR), San José de Lupuna town (LT), and San Pedro (SP). CAH site is located on both sides of the Iquitos-Nauta road between the 54th and 63rd kilometer (latitude 04˚13.785'S longitude 73˚276' W, located 2 hours by bus from Iquitos), and people are distributed among three rural villages in the southern part of San Juan district, La Habana (HA), Doce de Abril (DA) and Cahuide town (CT). STE site is a remote village located in the Nahuapa brook (04˚11′58.99″S, 74˚12′20.12″W), which can only be accessed by traveling 98 km by road from Iquitos to Nauta city (4 hours by bus), and then a 144 km river trip through the Marañon, Tigre and Nahuapa rivers (12 hours boat trip) [6]. A census conducted in March 2013 in this village identified 213 individuals. Villagers in the three study sites mainly work in agricultural activities, but in STE people regularly travel to trade and barter their products in Nauta (Information obtained by direct communication with the inhabitants of the area).

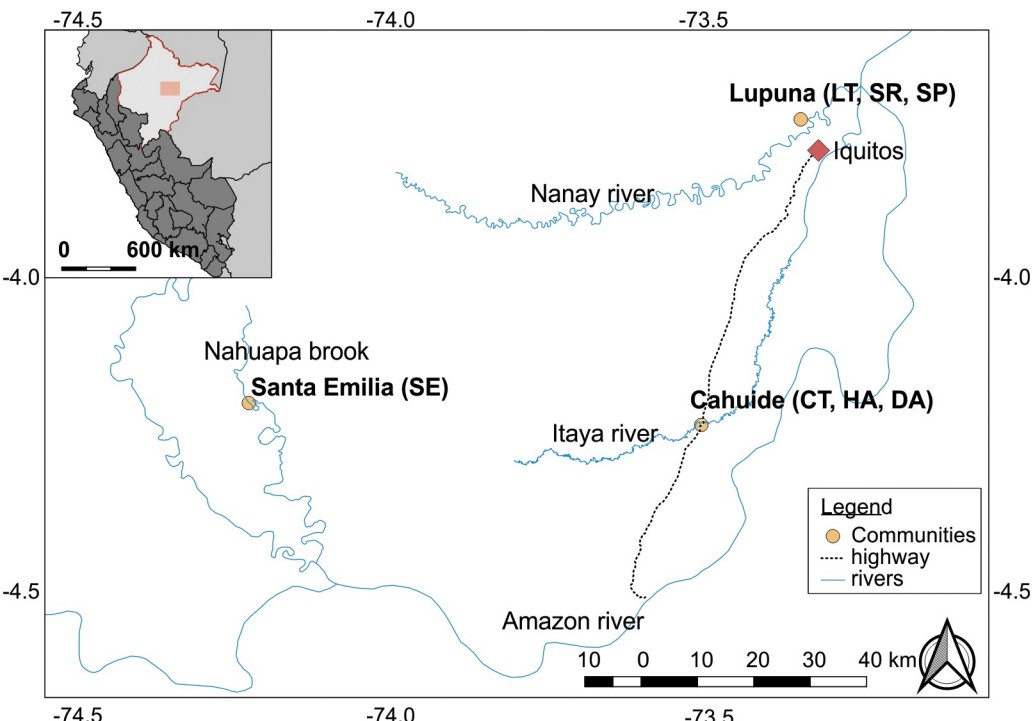

**Fig 1. Map of the study sites.** The map represents the three study areas included in this study. Lupuna (LUP) englobe three contiguous villages Santa Rita (SR), San José de Lupuna town (LT), and San Pedro (SP); Cahuide is composed by La Habana (HA), Doce de Abril (DA) and Cahuide town (CT); and Finally, Santa Emilia (STE) just include a town of the same name (SE). The Map was produced using QGIS Development Team 2.18.14 (2018), and the base map was obtained from The USGS Global Visualization Viewer GloVis (http://glovis.usgs.gov/), under CC BY 4.0.

## Study design and data collection

A three-year population-based, longitudinal cohort study was conducted in CAH and LUP from September 1st, 2012 and December 31st 2015, and a one-year cohort in STE from March 1st, 2013 to February 28th, 2014 (the STROBE Checklist in S1 Checklist). The study enrolled all available individuals who resided in the study sites and agreed to participate in the cohort providing written signed consent. Follow-up of study participants for the entire duration of the study was done by combining routine passive case detection (PCD) at the health posts and two active case detection (ACD) methods: monthly population screenings (mPS) and weekly active case detection of symptomatic individuals (wACDS). All participants were immediately screened for malaria by light microscopy regardless of the presence of symptoms. If the slide was positive, a health worker provided the anti-malarial treatment to the patient according to Peruvian national guidelines. Chloroquine plus Primaquine for *P. vivax* (CQ total 25 mg/kg over three days, and seven days PQ, 0.50 mg/kg/day which started simultaneously with CQ) and Artesunate-base combination therapy for *P. falciparum* and mixed infections (three days AS, 4 mg/Kg/day, and two days MQ, 12.5 mg/Kg/day which started one day after AS). Dried blood samples on filter paper from March, June, September, and December in CAH and LUP (from September 2012 to March 2015) were analyzed immediately by qPCR for parasite detection and quantification and by genotyping methods, while all filter paper samples from STE were analyzed at the end of the follow-up in June 2014.

## Molecular diagnosis and parasite quantification

Genomic DNA was isolated from ~6 mm$^2$ pieces of dried filter paper blood spots using QIAamp DNA Microkit (QIAGEN, Germany) following the manufacturer instructions, and the DNA was stored at -20˚C until its use in subsequent molecular methods. Molecular diagnosis by real-time PCR was performed following the modified protocol reported by Mangold, Manson [19] and using the following primers PL1473F18 [5'-TAA CgA ACg AgA TCT TAA-3'] and PL1679R18 [5'-gTT CCT CTA AgA AgC TTT-3']. The PCR reaction was performed at a final volume of 25 μL which included 12.5 μL of PerfeCTa SYBR Green FastMix (Quantabio, USA), 300 nM of each primer and 5 μL of DNA. The amplification was performed in a Real-Time PCR thermocycler CFX Connect TM Bio-Rad (Bio-Rad, USA), and the cycling conditions consisted of an initial denaturation at 95˚C for 2 minutes, followed by an amplification for 40 cycles of 20 seconds at 95˚C, 20 seconds at 52˚C, and 30 seconds at 68˚C with continuous image acquisition, then an extension at 68˚C for 3 minutes; and finally *P. vivax* positive infection was identified by melting temperature ($T_m$) between 77–77.5˚C.

The absolute quantification of the concentration of parasite DNA (in molecules per microliter) and parasite load (in parasites per microliter) present in each sample was made based on two standard curves elaborated from a plasmid pGEM-PV-18srRNA construct (from $2 \times 10^6$ mol/μL to 2 mol/μL) and a high concentration *P. vivax* sample (from $3 \times 10^3$ parasites/μL to 3 parasites/μL). The real-time qPCR used for diagnosis in this work has a limit of detection above 0.2 genomes per reaction (0.04 mol/μL), and because the subsequent PCRs are less sensitive than this qPCR, the samples were classified according to their DNA concentration. Only those *P. vivax* samples with a copy number greater than 24 mol/μL were diluted to 8 mol/μL and used directly in the amplification of markers used. Any sample that had a concentration between 24 mol/μL and 0.8 mol/μL was re-amplified by using the Illustra GenomiPhi V2 DNA Amplification kit. Samples below 0.8 mol/μL were discarded (S1 Fig).

## Microsatellite genotyping

Genotyping was carried out using a panel of 16 microsatellites, 9 of them reported in previous studies conducted in the Peruvian and Brazilian Amazon [20, 21]. Additionally, 7 microsatellite markers were used in physical linkage on chromosomes 2 and 14 with the aim of improving discrimination between identity by state and identity by descent. These new markers were identified and designed based on the genome of Sal1 referential strain, they have also been shown to be polymorphic, allow to determine the multiplicity of infection, and establish the genetic relationships between primary infections and recurrence episodes [22].

Forward primers were labeled with a fluorophore (6-FAM, VIC, NED, or PET) to analyze its product by capillary electrophoresis in the ABI PRISM 3100 system. The amplification was performed using one unit of the AccuStart II DNA polymerase enzyme, 2.5 μL of PCR Buffer, 1.5 mM of MgCl2, 200–400 nM (depending on the primer) of each of the primers, and 2.5 μL of DNA. The reaction consisted of an initial denaturation at 95 ˚C for 5 minutes, followed by 40 cycles of 30 seconds at 94 ˚C, 30 seconds at 55–67 ˚C (depending on the primer), and 30 seconds at 72 ˚C; and finally, a final extension of 15 minutes at 72 ˚C. The list of primers for the amplification of microsatellites against *P. vivax* is shown in S1 Table.

The amplified samples were analyzed by capillary electrophoresis in the ABI PRISM 3100 system, using the GeneScan 500 LIZ base pair marker. The chromatograms were analyzed using the GeneMapper Software Version 4.0 (Applied Biosystems). To confirm the accuracy of the equipment in the determination of the alleles, 4 samples (three mono-infections, and one mock polyclonal infection) were genotyped 12 times (one per each analyzed plate) by the set of 16 microsatellites. The genotyping precision was defined based on the standard deviation of

the allele size estimate for each locus. The recommended standard deviation for the equipment is 0.5 base pairs [23], so this value was taken as a reference to define the accuracy of the size estimation in base pairs of each allele.

Any peak with a relative fluorescence greater than 50 RFUs (relative fluorescence units) was considered a true allele. Those samples that presented more than one peak or allele at a given loci were considered as polyclonal samples, that means, there is the presence of more than one clone of parasites in the sample. Previous studies considered as alleles those secondary peaks that were greater than one third [11, 20, 21, 24] or one-quarter [25, 26] of the main peak. However, the parasite density of the different parasite clones within the same infection is variable and the secondary clones may be in a proportion less than the above-mentioned threshold. Additionally, a study that used amplicon deep sequencing found more parasite clones compared to the use of microsatellite markers [27]. Therefore, to avoid underestimating the proportion of polyclonal infections, secondary peaks were considered as alleles if 1) they were one fifth of the size of the main peak, 2) their heights were more than 50 RFUs, 3) their estimated allele sizes coincide with alleles detected on monoclonal samples and 4) if the height of the peak was greater than the expected stutter peak. In polyclonal samples, only the main peak was used to define the haplotype. A threshold of at least 12 amplified loci (75%) was considered for each sample, so genetic analyzes was based only on those samples and on markers with less than 10% of null alleles in the whole population (S1 Fig).

## Genetic diversity and population structure

Microsatellite results were entered into an Excel database in GenAlex format, where the code of each allele was the size in base pairs of the PCR product. The haplotypic and epidemiological information of each genotyped sample are presented in S2 and S3 Tables. Analyzes of haplotypic and genetic diversities, and linkage disequilibrium were made using the *"poppr"* package [28, 29], while population mutation rate (*Theta* or $\theta$) was made using the *"pegas"* package [30], both developed in the R v3.3.3 program, and the description of all these genetic indicators was carried out at geographic (by village and study area) and temporal (by month and season) level. Haplotypic diversity was defined based on the richness of genotypes (*MLG* or number of multilocus genotypes found), and the Simpson diversity index (*1—D*). Genetic diversity was measured based on the expected heterozygosity ($H_{exp}$), where $H_{exp}$ is defined as the probability of choosing two random alleles from a data set [31], while the population mutation rate was measured based on the expected homozygosity (*Theta$_h$* or $\theta_h$) as described by Kimmel, Chakraborty [32]. To determine if there are significant differences in $H_{exp}$ between the different comparison groups (for example between villages, or seasons), a Monte Carlo test was performed with 10,000 haplotype resampling, implemented in the *"adegenet"* package in R v3.3.3 [33]. Further, both the total variance of the expected heterozygosity (caused by the intra and inter locus variance), and the intra locus variance (variance caused by the sampling process) were reported [31]. As an indirect measure of inbreeding, the Standardized Association Index ($I_A^S$) was estimated from single haplotypes (to avoid false inbreeding due to clonal propagation) for each population of interest, and 10,000 allele permutations were used to determine their statistical significance, where the null hypothesis was that the $I_A^S$ value is equal to zero [34, 35].

The degree of population differentiation was measured by calculating the $G''_{ST}$ proposed by Jost [36] and implemented in the R package *"mmod"* [37], while the evaluation of the source of population differentiation was made using the analysis of molecular variance (AMOVA) using the *"ade4"* R package [38], for which two hierarchical levels were used (study area and village),

and the statistical significance of the fixation indexes 'φ' (phi) was calculated using 10,000 permutations.

A Bayesian model implemented in the program STRUCTURE v2.3 [39] was used to determine the number of populations or genetic clusters present in the study areas, to assign each infection to a specific genetic cluster, and to describe the cryptic subpopulation structure present in each area and time of sampling. A linked model with admixture was used with 20 replicates for each value of k (from 1 to 15), and a burn-in period of 50,000, followed by 200,000 iterations of Monte Carlo Markov chains [40]. The genetic distance between the loci (in centimorgans) was defined based on the meiotic recombination reported in *P. falciparum* [41], for which the physical distance in base pairs (bp) between contiguous loci was divided by 17Kpb/cM. To obtain the optimal number of genetic populations, the analysis of the second order rate of change of the marginal likelihood (*ΔΔK*) described by Evanno, Regnaut [42], was performed using the R package *"pophelper"* [43]. The average ancestry coefficient of the 20 replicas made for each infection was obtained using CLUMPP version 1.1.2 [44]. Genetic diversity ($H_{exp}$) and its temporal change were also described for each cluster, and to evaluate genetic relatedness of genotypes within and between each genetic cluster, a minimum spanning network (MSN) was calculated under Bruvo's distance [45] based on a stepwise mutation model for microsatellite data in *'poppR'* package [29].

Recent demographic changes (bottleneck or recent expansion of population size) in the parasite population were assessed by measuring the excess or deficit of heterozygosity with the software BOTTLENECK v1.2.02 [46, 47]. The Wilcoxon's test was used to evaluate for heterozygosity excess or deficit, and the expected heterozygosity in an equilibrium population was estimated using 10, 000 iterations and assuming three different mutation models: the strict one-step stepwise mutation model (SMM), the infinite allele model (IAM), and the two-phase model (TPM). For TPM the proportion of SMM was 85% with a variance of 30. These demographic changes were tested by season in each study area, by parasite clusters in each study area, and by parasite cluster per season in each study area.

Any new infection of *P. vivax* detected by qPCR in individuals who already had a previous infection during the follow-up period was defined as recurrent infection, regardless of the cause of the recurrence (new infection or reinfection, relapse, or recrudescence). Recurrent infections were classified as homologous or heterologous according to its membership to the genetic clusters. Thus, homologous recurrent episodes were infections caused by a parasite belonging to the same genetic cluster than the previous infections; while heterologous recurrent episodes were infections caused by parasites belonging to a different genetic cluster than the previous infections. In this way, the dynamic of parasite clones replacement, and changes in parasite density were analyzed in individuals with recurrent episodes. Statistical differences in parasite density among episodes were assessed by a Wilcoxon signed-rank test, while differences in parasite density among study areas and between homologous and heterologous infections were evaluated using the Mann-Whitney U test.

## Results

The percentage of amplification of the microsatellite markers used decreased with respect to the initial DNA concentration of the sample. Samples with an initial concentration greater than 8 mol/μL (70 parasites/μL) had a percentage of amplification greater than 98%; whereas, samples with initial concentrations between 1.5–8 (15–70 parasites/μL), and 0.8–1.5 mol/μL (7–15 parasites/μL) had amplification percentages above 64% (average: 73%) and 40% (average: 57%) respectively. None sample below the concentration of 0.8 mol/μL (7 parasites/μL) could be successfully amplified for more than 40% of the total markers used, which caused

sub-microscopic infections to represent only 39% of the total of successfully genotyped samples (samples with more than 75% of amplified loci).

On the other hand, the maximum standard deviation of the size assigned to the genotyped alleles was 0.1977 (coefficient of variation less than 0.05%), showing a high precision in the estimation of the size of the PCR fragments; and also only 0.18% (1/560) of the positive controls and 3.75% (9/240) of the negative controls showed amplification of alleles not corresponding to their genotype (in the case of negative controls they should not show amplification) during the processing of the samples. In addition, the assignment of primary and secondary alleles in polyclonal samples was consistent when the main clone was in a proportion of 2 to 1 (95.65%) and 3 to 1 (100%) with respect to its secondary clone. When the proportion was 1/1, in 56% of the cases one of the alleles remained the main allele. However, the ratio between the heights of the secondary and main peaks in the control polyclonal sample was in a range of one to one point five and one to five, and for that reason, a secondary peak was considered when its height was above one fifth of the main peak.

## Genetic diversity

Three thousand two hundred sixty-four *P. vivax* monoinfections were detected by qPCR during the period of the study. Of these, seven hundred seventy-seven *P. vivax* mono-infections were genotyped with 12 or more microsatellites markers (366 in LUP, 274 in CAH, and 137 in STE) (S1 Fig). Four hundred ninety-nine unique haplotypes (232 in LUP, 165 in CAH, and 111 in STE) were found from September 2012 to March 2015. Three unique haplotypes were shared between LUP and CAH, one between CAH and STE, one between LUP and STE, and two among the three areas, and only one out of the 7 shared haplotypes had a high frequency in the three study areas (14 in LUP, 41 in CAH, and 9 in STE).

In the study period, the total proportion of polyclonal infections was 0.256 (0.227–0.288), and the genotypic ($1—D$) and genetic diversity ($H_{exp}$) was 0.986 and 0.644 ($\pm$ 0.0005), respectively. The population mutation rate or $\theta_h$ was 1.939 ($\pm$ 0.03), and a significant Linkage disequilibrium was seen in the whole population ($I_A^S$ 0.145, p < 0.001) (all genotypic and genetic diversity index by study area and village are presented in Table 1).

LUP showed the lowest proportion of polyclonal infections with 0.197 (0.159–0.241), CAH had a proportion of 0.263 (0.214–0.3180), and STE showed the highest proportion with 0.401 (0.323–0.485). No differences in genotypic diversity were found among study areas, but $H_{exp}$ was lower in LUP (0.544 $\pm$ 0.0012) than in CAH (0.591 $\pm$ 0.0015) and STE (0.596 $\pm$ 0.0028) (p < 0.001 and 0.01, respectively), and no significant differences were found between CAH and STE. The population mutation rate ($\theta_h$) showed a similar pattern than $H_{exp}$, being LUP the lowest with 1.100 ($\pm$ 0.02) and STE the highest with 1.938 ($\pm$ 0.038); CAH has a $\theta_h$ of 1.617 ($\pm$ 0.038). Linkage disequilibrium ($I_A^S$) was significant in all study areas and in all villages (p < 0.001) (Table 1), however, its magnitude differs in every geographic location. The $I_A^S$ was 0.220, 0.203 and 0.181 in LUP, CAH and STE respectively.

Among villages, the proportion of polyclonal infections was similar in LUP; while in CAH, CT (0.297) and HA (0.269) showed higher proportion than DA (0.184). SP showed the highest $H_{exp}$ in LUP (0.622 $\pm$ 0.0038, p < 0.005), and LT (0.528 $\pm$ 0.0039) and SR (0.482 $\pm$ 0.0036) showed significant differences in $H_{exp}$ against all villages from CAH and STE (p < 0.05). On the other side in CAH there were no significant differences in $H_{exp}$ among CT (0.589 $\pm$ 0.0023), DA (0.566 $\pm$ 0.0059), and HA (0.625 $\pm$ 0.014). LT showed the lowest $\theta_h$ (0.591 $\pm$ 0.034) among all villages; and in SP, $\theta_h$ (1.967 $\pm$ 0.056) was almost three times higher than its neighbor villages, LT (0.591 $\pm$ 0.034) and SR (0.785 $\pm$ 0.022). HA was the village with the highest $\theta_h$ (2.2 $\pm$ 0.125) and the village with the lowest number of genotyped samples too.

**Table 1. Haplotype and genetic diversity by study area and village.**

| Study area | Village | N | Polyclonal (CI 95%) | MLG | *1—D* | $H_{exp}$ (SE) | $Theta_h$ (SE) | $I_A{}^S$ |
|---|---|---|---|---|---|---|---|---|
| LUP | LT | 117 | 0.19 (0.13–0.27) | 89 | 0.982 | 0.528 ± 0.004 | 0.59 ± 0.03 | 0.184 |
| | SP | 107 | 0.18 (0.12–0.26) | 70 | 0.967 | 0.622 ± 0.004 | 1.97 ± 0.06 | 0.197 |
| | SR | 142 | 0.22 (0.16–0.29) | 95 | 0.952 | 0.482 ± 0.004 | 0.79 ± 0.02 | 0.264 |
| | **Total** | **366** | **0.20 (0.16–0.24)** | **232** | **0.975** | **0.544 ± 0.001** | **1.10 ± 0.02** | **0.220** |
| CAH | CT | 172 | 0.30 (0.23–0.37) | 109 | 0.958 | 0.589 ± 0.002 | 1.40 ± 0.05 | 0.181 |
| | DA | 76 | 0.18 (0.11–0.29) | 50 | 0.948 | 0.566 ± 0.006 | 1.60 ± 0.05 | 0.297 |
| | HA | 26 | 0.27 (0.14–0.46) | 23 | 0.944 | 0.625 ± 0.014 | 2.20 ± 0.12 | 0.164 |
| | **Total** | **274** | **0.26 (0.21–0.32)** | **165** | **0.962** | **0.591 ± 0.001** | **1.62 ± 0.04** | **0.203** |
| STE | **SE** | **137** | **0.40 (0.32–0.49)** | **111** | **0.986** | **0.596 ± 0.003** | **1.94 ± 0.04** | **0.181** |
| **Total** | | **777** | **0.26 (0.23–0.29)** | **499** | **0.986** | **0.644 ± 0.0004** | **1.94 ± 0.03** | **0.145** |

N: sample size; **Polyclonal**: proportion of poyclonal infections; **MLG**: number of multilocus genotype; *1-D*: Simpson diversity index; $H_{exp}$: expected heterozygosity; $Theta_h$: population mutation rate based on the expected homozygosity; $I_A{}^S$: standardized index of association, all p-value of $I_A{}^S$ were below 0.001 (H₀: $I_A{}^S$ = 0).

Linkage disequilibrium ($I_A{}^S$) was significant in all villages (p < 0.001) (Table 1), being strongest in SR (0.264) and DA (0.297) and below 0.2 but above 0.15 in all other villages.

The temporal pattern of all these metrics varied among geographic locations (Fig 2). For example, CAH showed a clear reduction of cases (from 213 to 61) and a subtle reduction of polyclonal infections (from 0.286 to 0.18) after June 2013 (Fig 2A and S4 Table), but there were no reduction in $H_{exp}$ and $θ_h$ after this month (Fig 2B and 2C). Indeed, $H_{exp}$ showed a slight monthly increase till December 2013, and then its behavior was fluctuating. In contrast, in LUP $H_{exp}$ and $θ_h$ showed a significant increment from March 2014 forward (p < 0.005) (Fig 2E and 2F). In fact, in LUP from September 2012 to December 2013, $H_{exp}$ was 0.495 (± 0.003), and $θ_h$ was 0.517 (± 0.02), while from March 2014 to March 2015 they increased to 0.568 (± 0.002) and 1.435 (± 0.03), respectively. This increment in parasite diversity matched with the increment of genotyped samples and polyclonal infections (from 0.166 to 0.217) (Fig 2D and S4 Table). On the other hand, in the study area of STE neither of the metrics previously described showed a temporal change (Fig 2G–2I). Before July 2013 the proportion of polyclonal infections, $H_{exp}$ and $θ_h$ were 0.433 (0.341–0.529), 0.597 (± 0.004), and 1.942 (± 0.04); and from August 2013 forward they were 0.303 (0.174–0.473), 0.59 (± 0.012), and 1.763 (± 0.078), respectively.

## Population differentiation and population structure

Pairwise genetic differentiation among study areas ranged from moderate (0.113) to high (0.204), being stronger between LUP and STE (0.204), and between LUP and CAH (0.18); and weaker between CAH and STE (0.113) (Fig 2J). These results are consistent with their geographic proximity. Among villages between study areas, genetic differentiation ranged from moderate (0.11) to very high (0.336), while among villages within study areas pairwise genetic

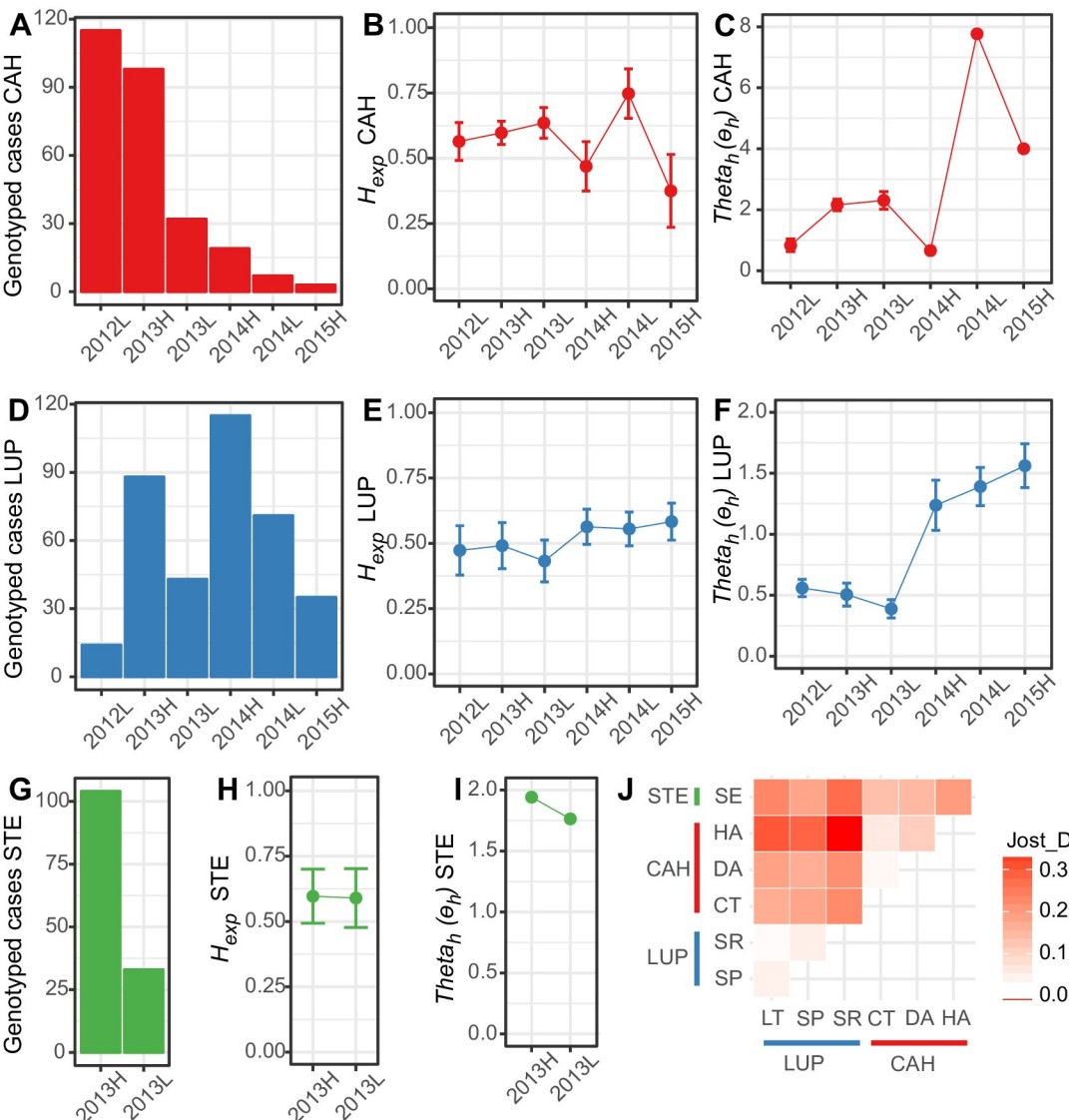

**Fig 2. Temporal change of the number of genotyping cases (A, D and G), the heterozygosity (B, E and H) and the population mutation rate or Theta (C, F and I) in Cahuide (Red), Lupuna (Blue), and Santa Emilia (Green).** Figure J is the graphic representation of the pairwise population differentiation proposed by Jost L. (2008).

differentiation was below 0.039 in most cases except between HA and DA where it was 0.085. These results were confirmed by the AMOVA analysis where just 1.38% of the genetic variation was explained by differences between villages within study areas, and 15.96% by differences between study areas. However, still, 82.66% was explained by differences within villages, which could be the sign of cryptic population structure or the presence of several private haplotypes (Table 2).

The analysis of the second order rate of change in the logarithmic marginal likelihood proposed by Evanno, Regnaut (42) suggested the presence of 3, 6, 11 or even 13 clusters. Despite the different numbers of probable clusters found, there was high consistency in the assignment of infections to an ancestral population when comparing the ancestry coefficient obtained

**Table 2. Analysis of molecular variance (AMOVA) by geographic location.**

| Source of variation | df | SS | MS | Sigma σ | % of variation | Phi (φ) | *p—value* |
|---|---|---|---|---|---|---|---|
| Between study areas | 2 | 23.61 | 11.80 | 0.04566 | 15.96 | 0.160 | 0.001 |
| Between villages within study areas | 4 | 2.47 | 0.62 | 0.00395 | 1.38 | 0.016 | 0.002 |
| Within villages | 770 | 182.07 | 0.24 | 0.23645 | 82.66 | 0.173 | 0.001 |
| Total | 776 | 208.14 | 0.27 | 0.28606 | 100 | | |

**df**: degree of freedom; **SS**: Sum of Squares; **MS**: Mean Sum of Squares.

assuming 3, 6, 11, and 13 ancestral populations (S2 Fig). In addition, consistency was observed between the genetic populations found and the clonal expansion observed through a minimum spanning network of haplotypes (S3 Fig), which suggests that the different levels of genetic subdivision are caused by clonal expansion events of the ancestral genetic populations of parasites within each geographic location or clonal expansion of the hybrids between those ancestral populations.

Assuming the presence of six clusters, clusters 1, 2 and 3 were the less diverse with an $H_{exp}$ of 0.221 (± 0.002), 0.174 (± 0.003), 0.132 (± 0.006), respectively (S5 Table shows the genetic diversity of each cluster by study area). The $H_{exp}$ in cluster 5 was 0.344 (± 0.008); and $H_{exp}$ was higher in cluster 4 (0.52 ± 0.002) and 6 (0.725 ± 0.002). In CAH, clusters 2, 3, 4, 5 and 6 were observed (Fig 3A), but clusters 2 and 5 were the only ones that persisted until March 2015 (Fig 3B). In fact, cluster 2 was the most frequent in CAH during the period September 2012—June 2013, and its diversity was one of the lowest of all the clusters present in this area (0.14 ± 0.005) (Fig 3C). In addition, all clusters but cluster 6 showed low diversity in CAH at the beginning of the study in 2012. Then the diversity of these clusters showed a slight increase during 2013; however, its $H_{exp}$ decreased again in 2014 (Fig 3C).

In LUP, the presence of clusters 1, 2, 3, 4 and 6 was observed (Fig 3A). Clusters 2 and 4 were observed throughout the study period, although the frequency of both was in decline, reaching only three and two cases in March 2015, respectively (Fig 3B). However, while the diversity of cluster 2 was always below 0.25 during the entire study period, the diversity of cluster 4 rose to more than 0.45 from the second half of 2013 to the end of 2014 and then decreased in 2015 (Fig 3D). On the other hand, clusters 1, 3, and 6 were observed in LUP just after March 2013 (Fig 3B). Indeed, cluster 1 remained the most frequent in this area from its entry (> 50% of cases) in 2013 till the end of the study (> 45%), and its diversity was always below 0.25 (Fig 3D). In contrast, cluster 6 (that always maintained a diversity above 0.65) frequency was noticeable from June 2014, representing up to 34% of cases in March 2015. In the case of cluster 3, this cluster was never above four cases in each season and its diversity was always less than 0.08.

In STE, clusters 2, 4, 5 and 6 were present throughout the year of follow-up (Fig 3A). Cluster 2 maintained low number of cases and low diversity (<0.25) throughout the follow-up period; while clusters 4 and 5 had low diversity at the beginning of 2013, which then increased to more than 0.5 at the end of the same year (Fig 3E). On the other hand, cluster 6 always maintained a diversity above 0.55. In addition, while in CAH and LUP the minimum spanning network showed clear indications of clonal expansion of each cluster in these areas, in STE only a small expansion of cluster 2 was evidenced (Fig 4).

## Demographic changes in the parasite populations

The SMM and TPM models showed a heterozygosity deficit throughout the different seasons in CAH and LUP (Table 3), which may be the product of a recent expansion of the population

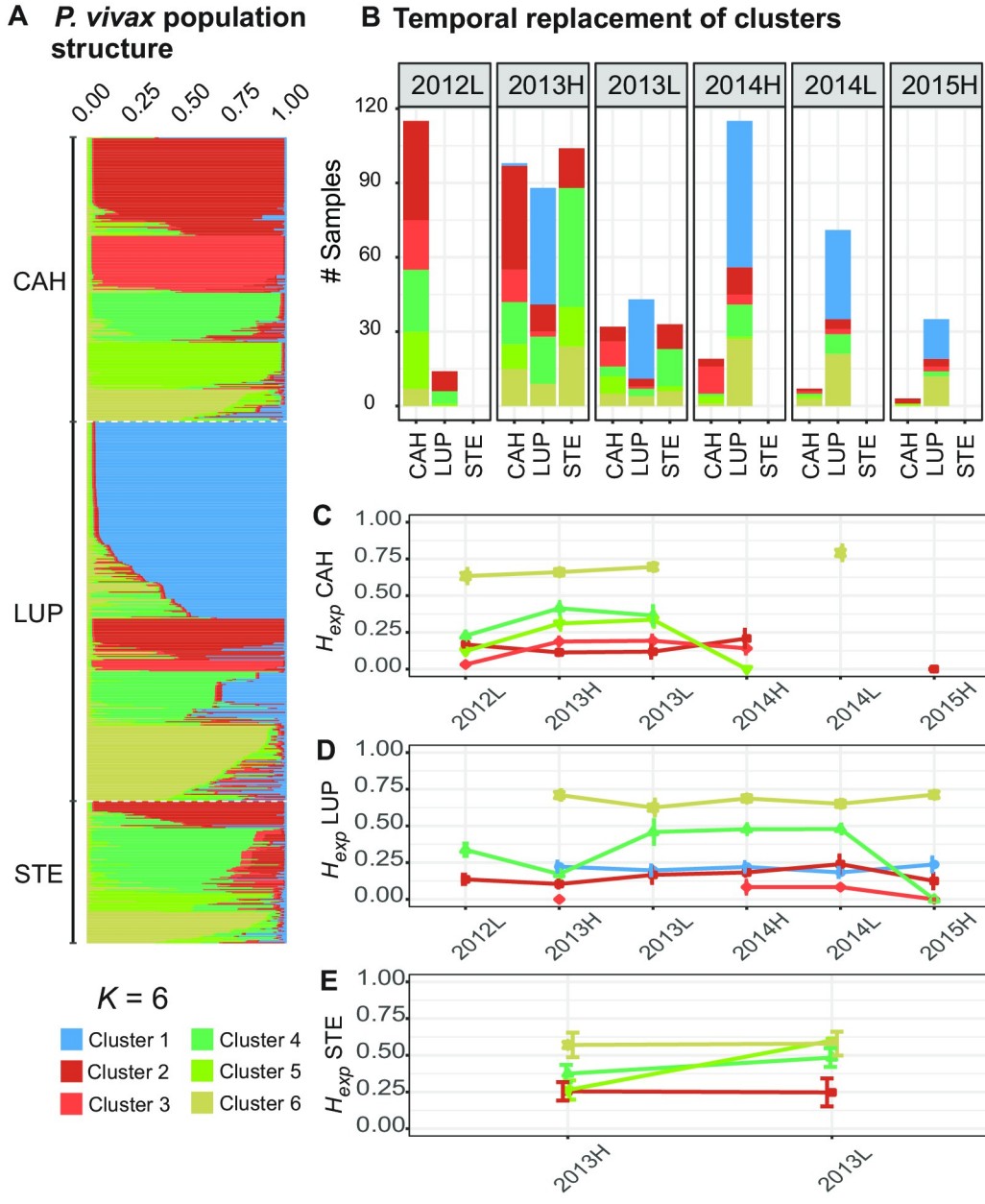

**Fig 3.** A) Parasite population structure in CAH, LUP and STE. B) Temporal replacement of parasite clusters in each study site. C-E) Temporal change in heterozygosity of each cluster by study area. Each color represents a cluster.

size or the recent influx of rare alleles from genetically distinct immigrants. This pattern was confirmed by analyzing each parasite subpopulation (clusters) in each study area over time (S6 and S7 Tables). Only STE showed an excess of heterozygosity (a recent bottleneck) under IAM, specifically for cluster 6 (for this cluster the excess was observed under the three models). This cluster also experienced an excess of heterozygosity under IAM in CAH and LUP, while all other clusters exhibited a deficit regardless of the model.

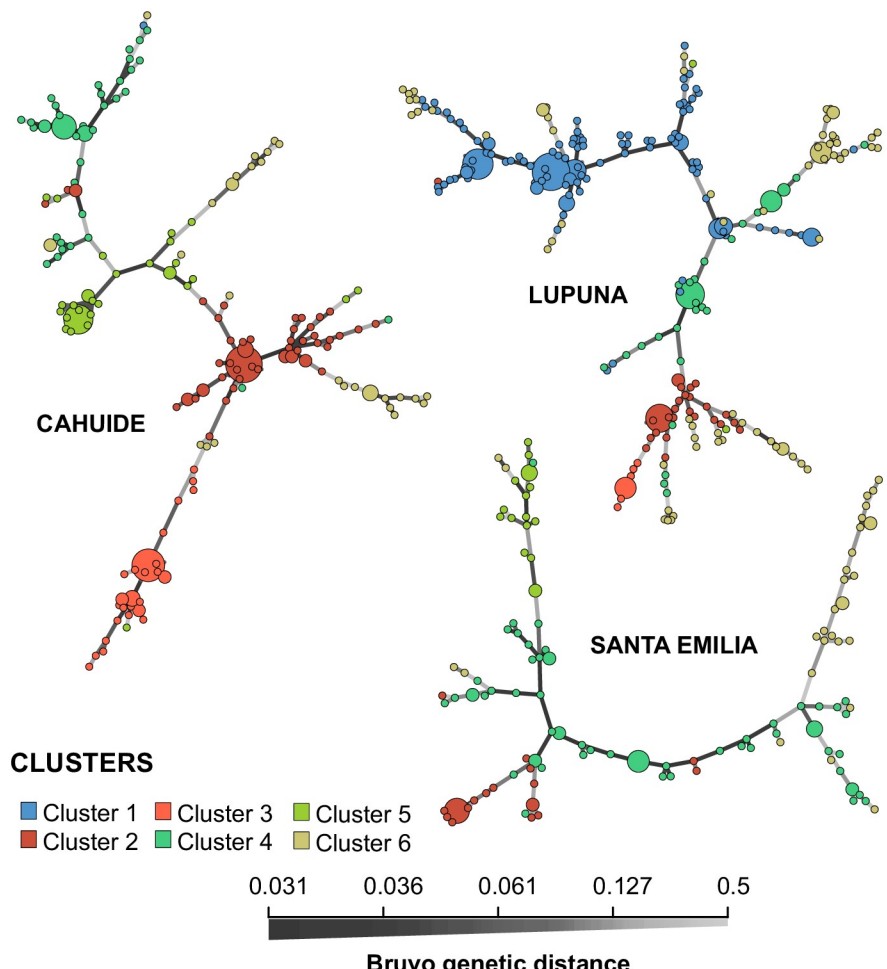

**Fig 4. Minimum spanning network of haplotypes based on Bruvo's distance with 16 microsatellite markers by study area.** Color denotes parasite clusters according to the population structure analysis. The circle size corresponds to the number of total individuals with the same haplotype. Branch darkness is proportional to inferred genetic distance between haplotypes. MSN was inferred assuming parsimony principle. The MSN is consistent with the cluster assignation made by Structure v2.3. Samples in each cluster from CAH and LUP are distributed in a few high-frequency haplotypes, and in haplotypes closely related to the high-frequency haplotypes; which is a sign of clonal expansion.

## Replacement of parasite genotypes and change in parasite load in individuals with recurrent infections

During the period of the study, 166 individuals had recurrent infections of *P. vivax*. One hundred twenty-seven out of them had 2 episodes, 27 had 3 episodes, 11 had 4 episodes, and only one individual in LUP had up to 6 episodes of *P. vivax*. Forty-two out of these 166 individuals were from CAH, 97 from LUP, and 27 from STE. Fig 5 presents the dynamic of replacement of parasite genotypes in individuals with recurrent episodes by each study area. In CAH, more than 50% of individuals whose previous episode of *P. vivax* was caused by clusters 2, 3, 5 or 6, presented a homologous recurrent episode, that means, an infection caused by a parasite belonging to the same genetic cluster than the previous infection. In contrast, in those individuals whose previous episode was caused by cluster 4, only 17% had a homologous recurrence. In LUP only individuals whose previous episode was caused by cluster 1 presented a high

**Table 3. Bottleneck analysis by study area and season based on three different mutation models.**

| Study area | Season | SMM | | | IAM | | | TPM | | |
|---|---|---|---|---|---|---|---|---|---|---|
| | | $p$ 1 tail | $p$ 2 tails | $H_{exp/eq}$ | $p$ 1 tail | $p$ 2 tails | $H_{exp/eq}$ | $p$ 1 tail | $p$ 2 tails | $H_{exp/eq}$ |
| CAH | 2012L | NS | < 0.01 | Deficit | NS | NS | | NS | < 0.05 | Deficit |
| | 2013H | NS | < 0.001 | Deficit | NS | NS | | NS | < 0.01 | Deficit |
| | After 2013L | NS | < 0.001 | Deficit | NS | NS | | NS | < 0.05 | Deficit |
| LUP | 2013H | NS | < 0.0001 | Deficit | NS | **NS** | | NS | < 0.0001 | Deficit |
| | 2013L | NS | < 0.0001 | Deficit | NS | < 0.001 | Deficit | NS | < 0.0001 | Deficit |
| | 2014H | NS | < 0.0001 | Deficit | NS | NS | | NS | < 0.001 | Deficit |
| | 2014L | NS | < 0.0001 | Deficit | NS | NS | | NS | < 0.001 | Deficit |
| | 2015H | NS | NS | | NS | NS | | NS | NS | |
| STE | 2013H | NS | < 0.05 | Deficit | < 0.001 | < 0.001 | Excess | NS | NS | |
| | 2013L | NS | NS | | < 0.05 | < 0.05 | Excess | NS | NS | |

$p$ **1 tail**: Wilcoxon's one tail p-value; $p$ **2 tails**: Wilcoxon's two tails p-value; $H_{exp/eq}$: behavior of the expected heterozygosity respect to the heterozygosity under population equilibrium; **SMM**: stepwise mutation model; **IAM**: infinite allele model; **TPM**: two phase model; **NS**: non-significant p-value.

percentage (> 45%) of homologous recurrences; whereas, in individuals with previous episodes caused by the other clusters present in this area, more than 30% of their recurrences were heterologous and caused by cluster 1. Finally, in STE over 55% of individuals presented homologous recurrences regardless of the origin of the genotype of the previous episode.

The standard curve based on a high concentration *P. vivax* sample had an efficiency of 90.10%, and its intercept in the y-axis was 37.10 of Cq, and these values were used to estimate the parasite density of the samples. The parasite densities of recurrent individuals were decreasing as the number of episodes increased, and this decrease was more evident and significant in LUP than in the other two study areas (Fig 6). Thus, in LUP a significant decrease (Wilcoxon signed-rank test, p <0.05) of parasite density was observed when comparing the parasite load of the 1st infection (median = 70.35 p/μL, IQR = 393.69 p/μL) with respect to the 2nd (median = 21.22 p/μL, IQR = 140.18 p/μL), 3rd (median = 17.2 p/μL, IQR = 40.39 p/μL) and 4th infection (median = 2.54 p/μL, IQR = 27.89 p/μL); and also, when comparing the 2nd with the 3rd infection (Wilcoxon signed-rank test, p <0.05). In addition, this decrease was surprisingly more marked in heterologous recurrences than in homologous recurrences. In fact, by subtracting the parasite density of the first infection with the density of the second episode (Fig 7), for both homologous and heterologous recurrences, significant differences were found (Mann-Whitney U test, p <0.05) between these two groups. Thus, it was observed that 70.97% of differences in the heterologous recurrences were greater than zero and had a median of 50.62 p/μL (IQR = 340.061 p/μL); while in the case of homologous recurrences only 37.14% of these differences were greater than zero and had a median of -10.12 p/μL (IQR = 435.46 p/μL), meaning that the parasitaemia of the second episode was even higher than the parasitaemia of the first episode.

A similar pattern was observed in CAH; however, only significant differences (Wilcoxon signed-rank test, p <0.01) were found when comparing parasite densities between the second (median = 158.16 p/μL, IQR = 644.65 p/μL) and third episode (median = 32.63 p/μL, IQR = 64.47 p/μL). In addition, although the subtraction of the parasite densities of the heterologous infections was greater than that of the homologous infections, these differences were not significant. On the other hand, STE presented significantly lower parasitic densities compared to the other two study areas, both when comparing the parasite density of all episodes (median = 8.37 p/μL, IQR = 57.83 p/μL, Mann-Whitney U test, p <0.05), and when comparing

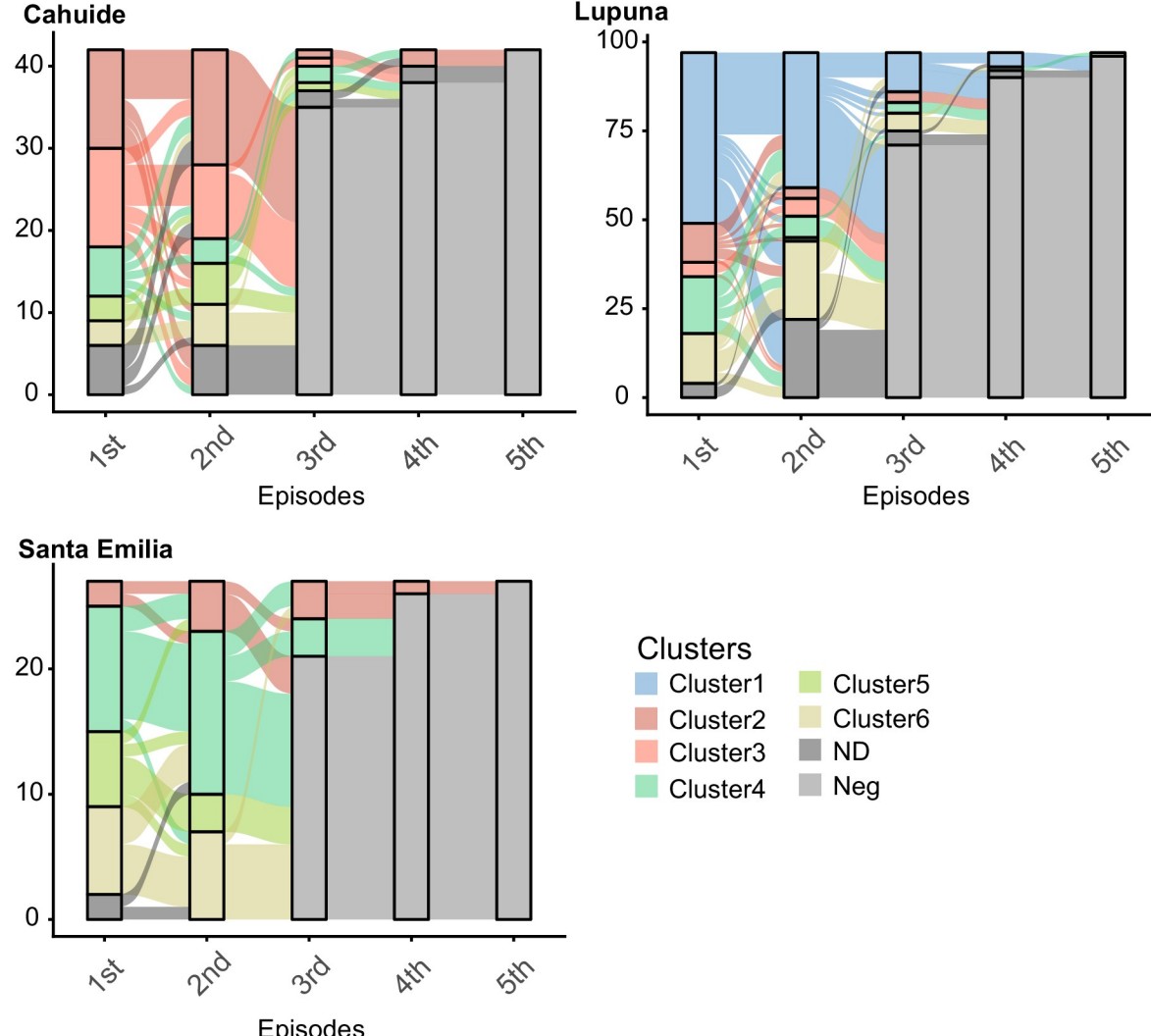

**Fig 5. An alluvial diagram that represents the replacement of parasite clones in individuals with recurrent episodes.** Episodes are in x-axis while the number of individuals with infection with a specific parasite clone is represented in the y-axis. Colors represent specific parasite clones or clusters, and cluster assignation was made by Structure v2.3.

first (median = 32.22 p/μL, IQR = 93.19 p/μL, Mann-Whitney U test, p <0.05) and second episodes (median = 7.49 p/μL, IQR = 29.43 p/μL, Mann-Whitney U test, p <0.05) individually. In addition, although a decrease in parasitic density was observed with the increase in the number of episodes, this decrease was not significant.

## Discussion

The data indicate the common occurrence of *P. vivax* gene flow supporting the hypothesis that human mobility connects geographically distant malaria transmission sites. There was a wide geographic distribution of at least six parasite genetic subpopulations (assuming *K* = 6) and the clonal expansion of one specific haplotype in the different study areas, which are signals of gene flow among them. Moreover, the AMOVA analysis showed that most genetic variation was explained within populations, which might suggest that the observed differentiation could

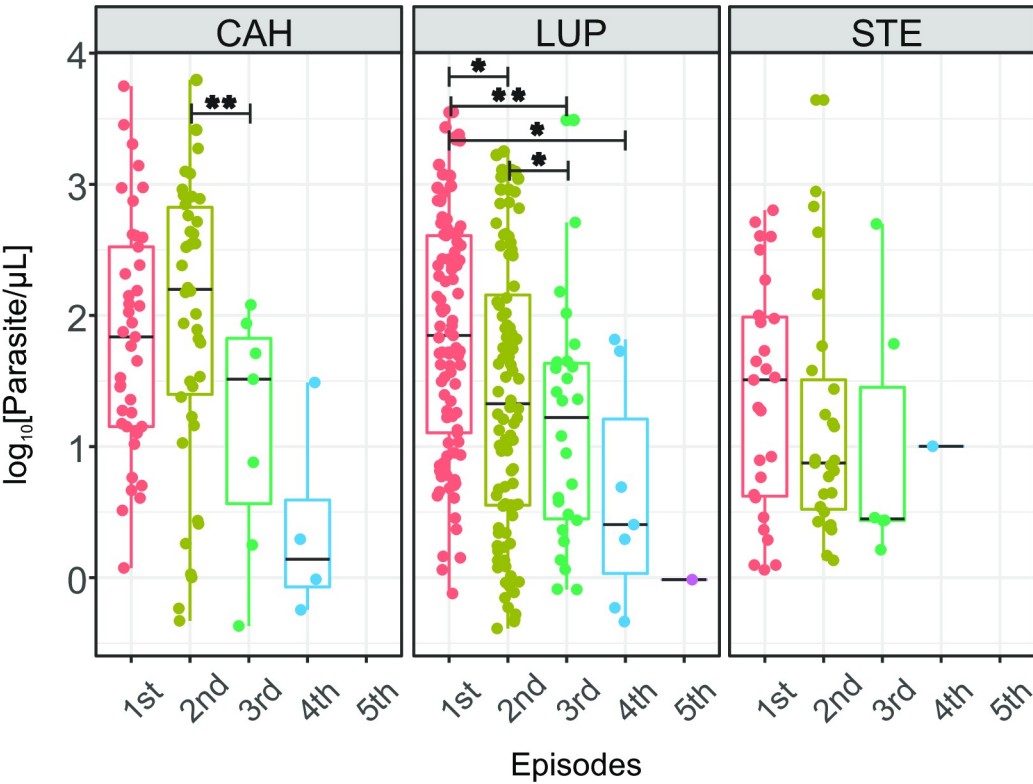

**Fig 6. Change of parasite load in individuals with recurrent episodes in each study area.** Episodes are in the x-axis while logarithm in base 10 of the parasite load of each infection is in the y-axis, and each dot represents the parasite load of the episode of one individual. Statistical differences in parasite load of the different episodes were determined by the Wilcoxon signed-rank test. *: p-value < 0.05, **: p-value <0.01.

be a sampling effect of alleles in low frequency more than geographic isolation. Similar patterns have been seen in previous studies in the Peruvian Amazon and in a recent study conducted in Colombia [11, 12], where the presence of a *P. vivax* malaria corridor that connect endemic areas and allow the persistence of parasite populations was proposed. A corridor is a group of populations connected in ways that enable gene flow among them; however, it does not necessarily involve long-range human travel. There could be areas acting as hubs that can maintain enough gene flow regionally allowing for recolonizations through time, even after several parasite "generations" [12]. The model seems to fit the observed patterns in the Peruvian Amazon where populations such as Iquitos, Nauta and Mazan could act as such hubs.

The temporal analysis presented here also showed the maintenance or increase in genetic diversity and proportion of polyclonal infections in LUP and STE, and also a deficit in the expected heterozygosity in LUP, which suggest a limited efficacy of the interventions applied in these sites to reduce the parasite population [48–50]. In LUP the introduction and rapid dispersal of cluster 1 was detected in March 2013, and this genetic cluster persisted until the end of the follow-up. The long distance plus the irregularity of river transport make STE a geographically isolated area, and because of that, until 2013 (when this study began) there were no interventions to control malaria in this site. This lack of control was reflected in the lower inbreeding, the higher genetic diversity and proportion of polyclonal infections in the area with respect to CAH, LUP and other endemic regions in the Peruvian Amazon [11, 13, 24, 51]. The genetic diversity and proportion of polyclonal infections reported in this site were only

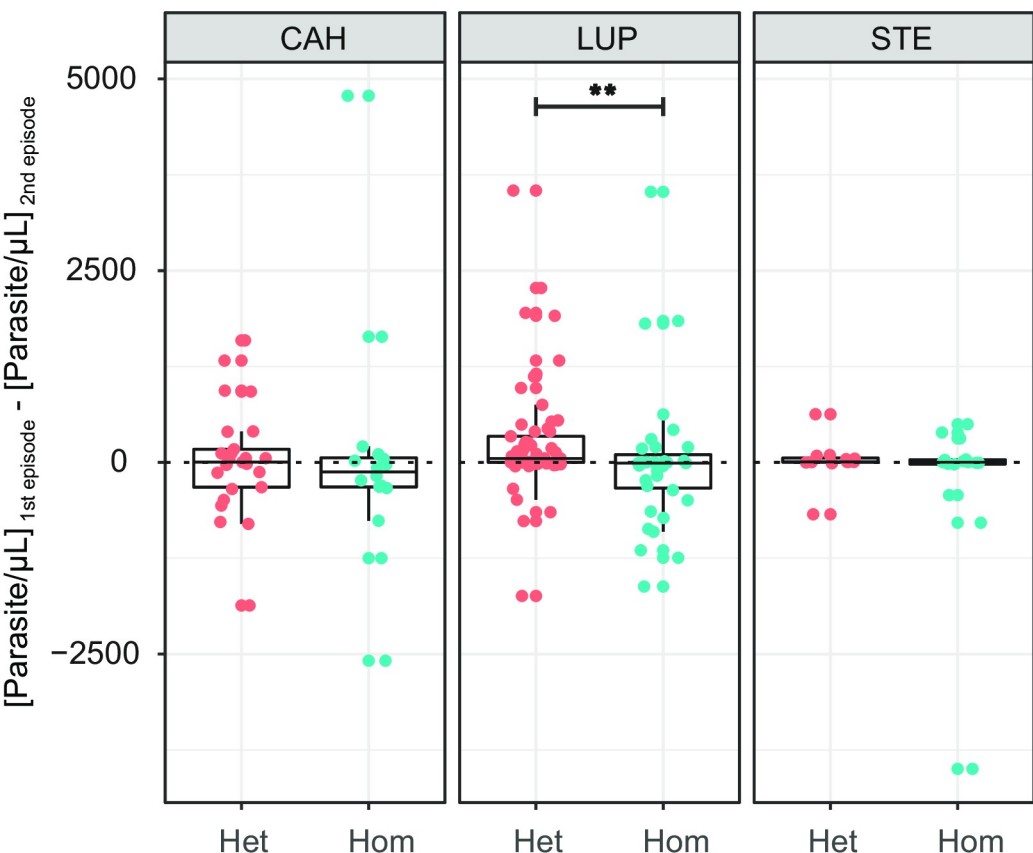

**Fig 7. Difference in the parasite density between first and second episodes in heterologous (Het) and Homologous (Hom) recurrences.** Difference in the parasite load of each individual is in y-axis. Statistical differences between Het and Hom episodes were determined by the Mann-Whitney U test. **: p-value <0.01.

comparable to that reported in the district of Mazan at the northwest of Iquitos [13], an area characterized by a transmission related to high human mobility through its basins [10]. This highlights the importance of the occupational mobility made by the settlers of STE to trade and barter their products in the village of Nauta.

In contrast to LUP and STE, the data suggest that in CAH the malaria cases depend on environmental changes that promote the presence of the vector such as flooding that occurred in 2012 [9]. Monthly population screenings (mPS), weekly active case detection of symptomatic individuals (wACDS) and several vector controls were provided in all sites but they were only effective here [6], where a significant reduction in the number of cases, the proportion of polyclonal infections, and the genetic diversity of the parasite populations were seen. Previous studies conducted in this area have characterized this site as an area highly vulnerable to the importation of malaria [11], and the reduction of cases observed in this site coincided with the reduction of the vector density since August 2013 [6]. So, the changes in diversity observed since 2014, might suggest that reported cases in CAH are originated in other neighbor communities in San Juan district or they could be the product of relapses or both.

Different patterns of change in parasite load and parasite clone replacement were observed in patients with recurrent infections among the three sites. In STE people with recurrent episodes control parasite density more efficiently than in the other two sites, showing low parasite densities even in the first episode recorded by this study (median = 32.22 p/µL, IQR = 93.19

p/μL). By contrast, a slow decrease in parasite load was observed in LUP and CAH. Furthermore, in LUP people with homologous recurrences, mainly with the recently introduced genetic cluster 1, did not control the parasite load of the recurrent infections efficiently. These different patterns in parasite load control (anti-parasite response) would be the result of differences in the distribution of genetic variability and exposure among malaria transmission sites in the Peruvian Amazon. Nonetheless, genetic variation at neutral markers like microsatellites do not necessarily represent the antigenic variability [52, 53], so investigation of the antigenic variability and the examination of the immune response over this population would be required to confirm these findings.

This study has some limitations. The low sensitivity of the genotyping methods compared to the sensitivity of the real-time PCR used for the diagnosis caused that just 34% of submicroscopic infections were successfully genotyped. It is necessary to use better methods of sample collection and develop and implement more sensitive genotyping methods to allow the study of this group of infections adequately. Another limitation is the lack of spatial and temporal resolution of this and other population genetic studies [12]. In CAH and LUP only samples from March, June, September, and December of each year of follow-up were diagnosed by real-time PCR and genotyped, which may have generated the loss of many submicroscopic infections in the intermediate months, and especially during the high transmission seasons. Even though the clonal propagation of one specific parasite haplotype was found in the three sites, people hardly ever travel among them, so parasite dispersion is not direct among these areas. Instead of that, parasite dispersion seems to occur in a progressive process through displacement to neighbor villages and economic attractors like Iquitos, Nauta and Mazan. For that reason, in order to describe the dynamic of the parasite dispersion and the identification of the contact zones that promote this parasite interchanging [12], future investigations require a suitable temporal and spatial sampling across these spatially connected endemic areas (malaria corridors).

The work presented here emphasizes the role of inter-community human mobility [54] that in combination with other factors like relapses and asymptomatic infections allows the presence of malaria corridors [12]. These corridors enable the persistence of the disease and may cause the reintroduction of parasites in areas where it has already been controlled (which would be called a "rescue effect"). This complex dynamic of parasite dispersion and distribution observed in the Peruvian Amazon must be considered in the design of control strategies so that these are focused and synchronized in an appropriate way with the dynamics of the transmission, making the chosen strategies more cost-effective.

## Supporting information

**S1 Checklist. STROBE checklist.**
(DOC)

**S1 Fig. Flow chart of individuals and sample selection for genotyping and population genetic analysis.** Samples with DNA concentration below 0.8 mol/μL were discarded because they were under PCR limit of detection. Samples with less than 75% of their genetic profile (less than 12 loci amplified) were also discarded. Selected samples were classified as monoclonal or polyclonal according to the number of alleles found per each locus.
(TIF)

**S2 Fig. Comparison of population structure pattern assuming 4 different values of ancestral populations, *K* = 3, *K* = 6, *K* = 11 and *K* = 13.** Each color represents an ancestral population, and samples have the same order in each graph.
(TIF)

**S3 Fig. Minimum spanning network of haplotypes based on Bruvo's distance with 16 microsatellite markers by *K* values.** Color denotes parasite clusters according to the population structure analysis for each *K*. The circle size corresponds to the number of total individuals with the same haplotype. Branch darkness is proportional to inferred genetic distance between haplotypes.
(TIF)

**S1 Table. List of primers for microsatellite amplification.**
(XLSX)

**S2 Table. Haplotypic information of genotyped samples.**
(XLSX)

**S3 Table. Epidemiological information of genotyped samples.**
(XLSX)

**S4 Table. Haplotypic and genetic diversity by season.**
(XLSX)

**S5 Table. Genetic diversity by cluster and study area.**
(XLSX)

**S6 Table. Bottleneck analysis by study area and parasite cluster based on three different mutation models.**
(XLSX)

**S7 Table. Bottleneck analysis by study area, parasite cluster and season based on three different mutation models.**
(XLSX)

## Acknowledgments

We would like to acknowledge and thanks all the field workers from the Amazonian ICEMR project who help in the enrollment and follow-up of the participants in this study. We are grateful for the support of the Direccion Regional de Salud (DIRESA, Iquitos, Loreto) and the communities of San Jose de Lupuna, Santa Rita, San Pedro, Cahuide, La Habana, Doce de Abril and Santa Emilia.

## Author Contributions

**Conceptualization:** Paulo Manrique, Gabriel Carrasco-Escobar, Mitchel Guzman-Guzman, Alejandro Llanos-Cuentas, Joseph M. Vinetz, Ananias A. Escalante, Dionicia Gamboa.

**Data curation:** Paulo Manrique, Julio Miranda-Alban, Gabriel Carrasco-Escobar.

**Formal analysis:** Paulo Manrique, Julio Miranda-Alban, Jhonatan Alarcon-Baldeon, Roberson Ramirez, Gabriel Carrasco-Escobar, Henry Herrera, Ananias A. Escalante.

**Funding acquisition:** Alejandro Llanos-Cuentas, Joseph M. Vinetz, Dionicia Gamboa.

**Investigation:** Paulo Manrique, Julio Miranda-Alban, Jhonatan Alarcon-Baldeon, Roberson Ramirez, Henry Herrera, Mitchel Guzman-Guzman, Angel Rosas-Aguirre, Ananias A. Escalante, Dionicia Gamboa.

**Methodology:** Paulo Manrique, Julio Miranda-Alban, Jhonatan Alarcon-Baldeon, Roberson Ramirez, Henry Herrera, Mitchel Guzman-Guzman.

**Project administration:** Joseph M. Vinetz, Dionicia Gamboa.

**Resources:** Mitchel Guzman-Guzman, Dionicia Gamboa.

**Software:** Paulo Manrique, Jhonatan Alarcon-Baldeon.

**Supervision:** Mitchel Guzman-Guzman, Joseph M. Vinetz, Ananias A. Escalante, Dionicia Gamboa.

**Validation:** Angel Rosas-Aguirre.

**Writing – original draft:** Paulo Manrique, Joseph M. Vinetz, Ananias A. Escalante, Dionicia Gamboa.

**Writing – review & editing:** Paulo Manrique, Julio Miranda-Alban, Jhonatan Alarcon-Baldeon, Angel Rosas-Aguirre, Alejandro Llanos-Cuentas, Joseph M. Vinetz, Ananias A. Escalante, Dionicia Gamboa.

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
