## [Decision Letter · Decision Letter 0]

11 Sep 2019

Dear Dr. Manrique:

Thank you very much for submitting your manuscript "Microsatellite analysis reveals connectivity among geographically distant transmission zones of Plasmodium vivax in the Peruvian Amazon: A critical barrier to regional malaria elimination" (PNTD-D-19-01028) for review by PLOS Neglected Tropical Diseases. Your manuscript was fully evaluated at the editorial level and by three independent peer reviewers. The reviewers appreciated the attention to an important topic but identified some aspects of the manuscript that should be improved.

We therefore ask you to modify the manuscript according to the review recommendations before we can consider your manuscript for acceptance. Your revisions should address the specific points made by each reviewer.

(1) A letter containing a detailed list of your responses to the review comments and a description of the changes you have made in the manuscript.

(2) Two versions of the manuscript: one with either highlights or tracked changes denoting where the text has been changed (uploaded as a "Revised Article with Changes Highlighted" file ); the other a clean version (uploaded as the article file).

(3) If available, a striking still image (a new image if one is available or an existing one from within your manuscript). If your manuscript is accepted for publication, this image may be featured on our website. Images should ideally be high resolution, eye-catching, single panel images; where one is available, please use 'add file' at the time of resubmission and select 'striking image' as the file type. 

Please provide a short caption, including credits, uploaded as a separate "Other" file. If your image is from someone other than yourself, please ensure that the artist has read and agreed to the terms and conditions of the Creative Commons Attribution License at http://journals.plos.org/plosntds/s/content-license (NOTE: we cannot publish copyrighted images). 

(4) Appropriate Figure Files 

Please remove all name and figure # text from your figure files upon submitting your revision. Please also take this time to check that your figures are of high resolution, which will improve both the editorial review process and help expedite your manuscript's publication should it be accepted. Please note that figures must have been originally created at 300dpi or higher. Do not manually increase the resolution of your files. For instructions on how to properly obtain high quality images, please review our Figure Guidelines, with examples at: http://journals.plos.org/plosntds/s/figures

While revising your submission, please upload your figure files to the Preflight Analysis and Conversion Engine (PACE) digital diagnostic tool, https://pacev2.apexcovantage.com/ PACE helps ensure that figures meet PLOS requirements. To use PACE, you must first register as a user. Then, login and navigate to the UPLOAD tab, where you will find detailed instructions on how to use the tool. If you encounter any issues or have any questions when using PACE, please email us at figures@plos.org.

We hope to receive your revised manuscript by the 11th October 2019. If you anticipate any delay in its return, we ask that you let us know the expected resubmission date by replying to this email.

To submit your revised files, please log in to https://www.editorialmanager.com/pntd/

Sincerely,

Donelly Andrew van Schalkwyk, Ph.D.

Guest Editor

Hans-Peter Fuehrer

Deputy Editor

Reviewer's Responses to Questions

Key Review Criteria Required for Acceptance?

Methods

-Are the objectives of the study clearly articulated with a clear testable hypothesis stated?

-Is the study design appropriate to address the stated objectives?

-Is the population clearly described and appropriate for the hypothesis being tested?

-Is the sample size sufficient to ensure adequate power to address the hypothesis being tested?

-Were correct statistical analysis used to support conclusions?

-Are there concerns about ethical or regulatory requirements being met?

Reviewer #1: Objectives and scopes of the study are clearly explained in the introduction. The study design is appropriate. The population is clearly described and the sample size sufficient to ensure adequate response. All thresholds used (allele definition, polyclonality definition, genotyping panel range) are clearly enounced in their material and methods section. 

Ethical and regulatory requirements are met.

Reviewer #2: The study objectives are fairly clearly. Specifically, the overarching aim is to use genetic data on P. vivax isolates collected in longitudinal cohort studies in three geographically distant sites in the Peruvian Amazon to assess the connectivity between sites and temporal changes in the dynamics of parasite transmission in this region. The authors intended to sue this information to derive insights into the major factors sustaining parasite transmission in the region.

The study design is appropriate to address the objectives.

The population is very comprehensively described. Limitations, such as the modest spatial and temporal resolution (which is a limitation in pretty much all genetic studies of this species to date) are appropriately reported in the discussion.

The sample size in each of the three sites is appropriate for the spatial analyses. The samples sizes for the analyses of changes in parasite density across recurrent infections are good for two sites (CAH and LUP) and modest for one (STE).

The analyses were carefully considered and are appropriate for the study objective.

There do not appear to be any ethical issues. The authors state that ethical approval was obtained from the relevant ethics committees, with patients providing informed written consent or assent for participation and blood collections.

Reviewer #3: The work by Manrique et al studies the population genetic structure of different transmission zones in the Peruvian Amazon and its change over the years, importation and clonal propagation. 

The objectives of this study are clear and the authors hypothesized no changes in genetic flow over the years due to high human mobility. The study design and sample size appear appropriate. The authors used molecular diagnosis and quantified the parasites to then genotype them using 16 microsatellite markers. The analysis of diversity and population structure was done through state of the art methods using R packages and specialized software. All ethical procedures were followed.

Results

-Does the analysis presented match the analysis plan?

-Are the results clearly and completely presented?

-Are the figures (Tables, Images) of sufficient quality for clarity?

Reviewer #1: The responses to the three above questions are appropriate and the manuscript meet these requirements.

Reviewer #2: The analysis presented matches that proposed in the analytical plan.

The results are clearly and comprehensively presented.

The figures are coherent and of suitable quality.

Reviewer #3: The results are clearly presented: The authors found 26% polyclonal infections, a large number of unique haplotypes, medium to high diversity and significant linkage disequilibrium. The authors characterized the changes in structure through time in the population and in recurrent infections. There were differences in diversity indexes levels among the communities and the genetic composition proportion of clusters varied between the study sites. Nevertheless, the same clusters were found in the different sites, indicating that human mobility connects the different study sites and confirming their hypothesis. 

Ther figures are in general clear and comprehensive.

Conclusions

-Are the conclusions supported by the data presented?

-Are the limitations of analysis clearly described?

-Do the authors discuss how these data can be helpful to advance our understanding of the topic under study?

-Is public health relevance addressed?

Reviewer #1: The manuscript meet the above requirements.

Reviewer #2: The conclusions are supported by the data presented.

Study limitations, including issues with genotyping low density infections, and limited spatial and temporal resolution (see comment above) are clearly reported.

The authors present a well considered interpretation of the results, highlighting how they inform on the main factors sustaining the P. vivax population in the Peruvian Amazon. A major factor appears to be a malaria 'corridor' underpinned by human mobility, connecting geographically distant populations and in doing so sustaining the persistence of the parasite populations across the region. As the authors conclude, these patterns are critical for consideration in local strategies to interrupt P. vivax transmission in the region.

As per the above statement, the public health relevance of the study findings is addressed.

Reviewer #3: The authors conclude that the mobility is a factor that leads to complex dynamics of parasite dispersion, which complicates the work of the control programs and needs to be taken into account in control strategies. Some limitations of this study were the difficulty of including a large number of submicroscopic samples in the study, the lack of spacial and temporal resolution and the difficulty of identifying the exact migration corridors.

Editorial and Data Presentation Modifications?

Reviewer #1: Lines 116-130: The authors present population movements whose intensity seems to be known. Are authors could give references and quantification if possible of these population movements?

Lines 206-209: Can the authors explicit how they define their threshold of 50RFUs for the allele definition. Have they realized any replications of the experiment for a sample set (in triplicates for example)?

Lines 211-212: The authors explicit that only genotyped samples with 75% or more of their genetic profile are considered. Do they also define a threshold of missing data per loci or all 12 selected loci were analyzed whatever their profile and succeed rate of amplification?

Lines 259-261 Could the author explain deeply their classification of recurrent infections? Does it was only based on the follow-up assuming that all new cases in a patient is a recurrent infection?

Lines 422-440: Was it possible (considering epidemiological data) to rule out the probability that a homologous infection was caused by a new infection rather than by a relapse? Likewise, studies show that the genetic profile of relapse may differ from that of the first infection, did the authors observe this?

Lines 480-482: Given the resilient nature of P. vivax parasites, did the authors also observed a specific genetic signature (bottleneck per se) supporting the limited efficacy of the interventions?

Reviewer #2: (No Response)

Reviewer #3: Some minor changes needed are:

- It is not clear what is the exact distance between the study locations and how they relate mobility wise. This should be addressed in the Methods section.

-Line 58: it is not clear what "introduction" is referred to

- Lines 140-141: it should be: from March 1st, 2013 to February 28th, 2014

- Line 207-208: The authors consider an allele any secondary peak higher that one fifht of the main peak. Why did they choose this measure, taking into account that other studies use more tan one third. Ana explanation or reference is needed. 

- Lines 315-329: Figure 2 is not correctly cited. The order of the parts of the figure is incorrect. Please re check the the order of the figure and change in the text or change the figure itself.

- Lines 466-467: The authors talk about 3 genetic subpopulations (assuming K=3). Is this correct? Shouldn't it be K=6?

- Line 537: should be emphasizes

Summary and General Comments

Reviewer #1: This paper of Manrique et al. underlines the impact of human mobility towards malaria elimination in the Peruvian Amazon. The authors used a microsatellite approach to infer the genetic diversity and population structure of Plasmodium vivax field isolates sampled during a 3-year (1-year for one site) longitudinal cohort studies in three geographically distant locations in the Peruvian Amazon. Objectives are well expounded in their introduction, clearly defining the scope of their study. All thresholds used (allele definition, polyclonality definition, genotyping panel range) are clearly enounced in their material and methods section. The article is well written and conclusions are well supported by the methodological approach, data and analyses. This study deepens knowledges on P. vivax in the Amazon Basin. This study is of very high interest to multiple disciplines of study including malaria control program of the country and Amazonian region in general as it brings informations on parasite population dynamic under or after the implementation of different elimination approaches.

Reviewer #2: The general approach used in the study is not particularly novel, but the findings add to an important body of evidence on the utility of parasite genetic data to inform on malaria control and intervention strategies. The data set is very comprehensive (at least for the spatial analyses)relative to other studies on P. vivax genetic diversity and in this regard, the evidence provided in the study is fairly robust.

Reviewer #3: This work represents an important advancement to the knowledge of P. vivax in the Amazon. It is highly relevant to it's field and to a neglected disease such as P. vivax. It provides clear information of the complexity of P. vivax populationon dynamics in the area.

PLOS authors have the option to publish the peer review history of their article (what does this mean?). If published, this will include your full peer review and any attached files.

Do you want your identity to be public for this peer review? For information about this choice, including consent withdrawal, please see our Privacy Policy.

Reviewer #1: No

Reviewer #2: No

Reviewer #3: No

---

## [Editor Report · Decision Letter 1]

25 Oct 2019

Dear Dr Manrique,

We are pleased to inform you that your manuscript, "Microsatellite analysis reveals connectivity among geographically distant transmission zones of Plasmodium vivax in the Peruvian Amazon: A critical barrier to regional malaria elimination", has been editorially accepted for publication at PLOS Neglected Tropical Diseases.

Before your manuscript can be formally accepted and sent to production you will need to complete our formatting changes, which you will receive in a follow up email. Please note: your manuscript will not be scheduled for publication until you have made the required changes.

IMPORTANT NOTES

* Copyediting and Author Proofs: To ensure prompt publication, your manuscript will NOT be subject to detailed copyediting and you will NOT receive a typeset proof for review. The corresponding author will have one final opportunity to correct any errors when sent the requests mentioned above. Please review this version of your manuscript for any errors.

* If you or your institution will be preparing press materials for this manuscript, please inform our press team in advance at plosntds@plos.org. If you need to know your paper's publication date for media purposes, you must coordinate with our press team, and your manuscript will remain under a strict press embargo until the publication date and time. PLOS NTDs may choose to issue a press release for your article. If there is anything that the journal should know, please get in touch.

*Now that your manuscript has been provisionally accepted, please log into EM and update your profile. Go to http://www.editorialmanager.com/pntd, log in, and click on the "Update My Information" link at the top of the page. Please update your user information to ensure an efficient production and billing process.

*Note to LaTeX users only - Our staff will ask you to upload a TEX file in addition to the PDF before the paper can be sent to typesetting, so please carefully review our Latex Guidelines [http://www.plosntds.org/static/latexGuidelines.action] in the meantime.

Best regards,

Donelly Andrew van Schalkwyk, Ph.D.

Guest Editor

Hans-Peter Fuehrer

Deputy Editor

---

## [Editor Report · Acceptance letter]

6 Nov 2019

Dear Mr. Manrique Valverde,

We are delighted to inform you that your manuscript, "Microsatellite analysis reveals connectivity among geographically distant transmission zones of Plasmodium vivax in the Peruvian Amazon: A critical barrier to regional malaria elimination," has been formally accepted for publication in PLOS Neglected Tropical Diseases.

Best regards,

Serap Aksoy

Editor-in-Chief

Shaden Kamhawi

Editor-in-Chief
